# Physical Culture and Sports as an Educational Basis of Students’ Healthy Physical Activities during and Post-Lockdown COVID-19 Restrictions

**DOI:** 10.3390/ijerph191811663

**Published:** 2022-09-16

**Authors:** Remigiusz Dróżdż, Marcin Pasek, Magdalena Zając, Mirosława Szark-Eckardt

**Affiliations:** 1Faculty of Physical Culture, Gdansk University of Physical Education and Sport, 80-336 Gdansk, Poland; 2Department of Special Pedagogy and Speech Therapy, Kazimierz Wielki University, 85-064 Bydgoszcz, Poland; 3Institute of Physical Education, Kazimierz Wielki University, 85-064 Bydgoszcz, Poland

**Keywords:** student physical activity, pandemic, remote teaching, traditional teaching

## Abstract

(1) Background: The SARS-CoV-2 virus manifests itself in many aspects of everyday life, among which one of the most frequently accentuated are changes in physical activity from the perspective of lockdown mobility limitations. The aim of this study was to assess the level of physical activity in groups of students with different levels of educational engagement in physical activity while taking into account the realities of remote and traditional education. (2) Methods: The study was conducted among 200 students of 3 majors with varying degrees of curriculum related to physical activity, including students of physical education (*n*-73), tourism and recreation (*n*-65) and students of a non-physical education major (*n*-62). The survey technique used was the short version of the International Physical Activity Questionnaire (IPAQ). (3) Results: PE students are the most engaged in physical activity compared to other study groups. This pattern is particularly true for high physical activity. An interesting observation is that, in the post-lockdown period, there was no statistically significant difference between the physical activity levels of PE students and TR students. Meanwhile, such a difference in favor of PE students was evident during the lockdown period. (4) Conclusions: Educational links to physical activity appear to be an important factor in stimulating this activity during times of epidemiological emergency.

## 1. Introduction

The concept of determinants of human health known as Lalonde’s fields, popularized in the 1970s, include categories with significant interrelations. Such interrelationships can be observed between, among others, lifestyle, which determines about 53% of health, and the environment (represented in these studies by coronavirus), which determines about 21% of individual well-being [1]. One of the key elements of a healthy lifestyle is systematically undertaken physical activity [2], since the turn of the 20th and 21st centuries treated together with dietary habits as an integral part of a healthy attitude [3].

The physical activity levels of young adults are often unsatisfactory [4,5]. The aim of physical education is to prepare the young person to participate in physical culture throughout life. During primary and secondary school, the conditions for participation in physical education classes are standardized in terms of the core curriculum. In contrast, the period of study in the Polish education system is not uniformly organized in this respect. Often individual universities interpret the obligation for students to take physical education classes quite freely. In universities whose didactic programs are not related to physical culture, the number of compulsory hours of physical education is usually 30 h delivered in one semester. In contrast, there are 135 h of physical education in the current curriculum for tourism and recreation studies. Meanwhile, physical education students in the study population have 700 h of compulsory physical activity in their study program. These significant differences may largely determine the level of daily physical activity expressed in MET units (metabolic equivalent of work). However, the organization of studies has changed in the last two years due to the restrictions accompanying the COVID-19 pandemic. The implementation of a significant number of teaching hours in an online format invites the question of to what extent the introduced restrictions regulate the level of daily physical activity of students of different specializations. Indeed, it is debatable whether the mobility restrictions resulting from COVID-19 will have the greatest negative impact on the activity levels of physical education students. In this view, the coronavirus is considered a direct health threat but also an indirect threat through restrictions aimed at limiting human physical activity [6].

Meanwhile, physical exercise has a role in increasing the activity of macrophages—immune cells that are one of the first lines of defense against infection. They are responsible for engulfing and degrading bacteria and inactivated viruses. Among the cells of the second line of defense, there are lymphocytes, whose increased number is also associated with physical activity. One of its effects is an increase in the number of antibodies accumulated in the blood and secreted on the surface of mucous membranes. These antibodies are responsible for blocking viral entry, further enabling the arrest of an already developing infection by activating the destruction of pathogens and infected cells [7].

The relationship between the degree of epidemiological risk and the level of declared physical activity is an example of the role of various environmental factors in shaping human physical activity, which has been extensively analyzed in the world literature. So far, the issues of physical exercise at extremely varying ambient temperatures [8], reduced atmospheric pressure [9], contact with plant aerosols [10] or anthropogenic air pollutants [11] have attracted interest.

The level of physical activity is a positive measure of population health [12]. The choice of the method of physical activity assessment depends mainly on the aim and scope of the study, understood as the size of the study group [13]. In cross-sectional and longitudinal studies, mainly questionnaire-based tests are used. In addition to the short time to perform, their additional advantages are that they do not require laboratory conditions and can be used regardless of age, gender and health status. Although technically the collection of research material is not troublesome, a comparison of the physical activity of different populations, and especially the determination of factors conditioning its level, is difficult. The problem is constituted by conceptual and terminological differences, cultural differences and many other modifying factors that are difficult to take into account [14,15,16]. Despite this, more and more research centers around the world are involved in cross-sectional studies of physical activity [17,18,19]. For at least 20 years, many international research communities have studied this subject, including the European Health Interview Survey—EUROHIS and the European Physical Activity Surveyance System—EUPASS but the creators of the European Social Survey particularly recommend the International Physical Activity Questionnaire—IPAQ [20]. It is even described as the only real method possible to use in studies of large populations [21].

The aim of this study was to assess the level of weekly physical activity of those studying physical education, tourism and recreation and non-physical activity specializations during and after the restrictions accompanying the COVID-19 pandemic were lifted.

The Polish reality regarding online teaching was related to the complete abandonment of physical activity during class time. The time allocated to physical education was devoted to the analysis of historical or methodological issues. In terms of structure, the classes were transformed from typically practical to theoretical. Thus, physical activity was almost completely eliminated during this period. A question worth exploring, therefore, is to what extent the students of the respective specializations were motivated to be physically active during the lockdown period and whether this level of activity clearly changed in the post-lockdown period.

## 2. Materials and Methods

The study was conducted among 200 students (97 males and 103 females) of 3 fields of study with varying relationships to physical activity. Physical education students constituted a group of 73 (38 males and 35 females). The study also included 65 tourism and recreation students (32 males and 33 females) and 62 non-physical education students (27 males and 35 females). The mean age of the study population was 21.70 ± 1.59, including physical education students at 21.34 ± 1.71, tourism and recreation students at 21.15 ± 1.35 and other students at 22.68 ± 1.20. The entire study group were of Polish nationality and originated from two universities located in Gdańsk.

The physical activity level of the population was assessed using a diagnostic survey method. The survey technique used a tool in the form of a short version of the International Physical Activity Questionnaire—IPAQ [14]. It expresses physical activity in units of MET-min/week, which makes it easy to classify respondents into one of three activity categories: insufficient (less than 600), sufficient (600–1500 or 600–3000) or high (more than 1500 or 3000 MET-min/week). In the international literature, these categories are usually described as low, moderate and high activity [22]. The short version of the questionnaire includes 7 questions on all types of physical activity related to daily life, work and leisure. Activities performed at work; at home; in the home environment; in moving from place to place; and in leisure time devoted to recreation, exercise, or sport were considered. Respondents in the lockdown period completed the questionnaire online, while in the post-lockdown period they completed a traditional paper version. The survey was carried out in homes during the lockdown period, in May 2020, and the post-lockdown survey—after students return to traditional university education—in October 2021.

It was initially hypothesized that students of physical education are more physically active than students of tourism and recreation and even more than students of non-tourism majors. This general assumption was specified in the form of five specific research hypotheses:

**H1**:
*The level of total physical activity calculated in METs and minutes per week in each of the 3 groups is higher during traditional teaching compared to the level of activity during remote teaching;*


**H2**:
*During both remote and traditional teaching, higher levels of total physical activity calculated in METs and minutes are shown by physical education students compared to tourism and recreation students;*


**H3**:
*During both remote and traditional teaching, higher levels of total physical activity calculated in METs and minutes are shown by tourism and recreation students compared to non-physical culture students;*


**H4**:
*Physical education students, regardless of gender, devote more attention per week to intensive physical activity during lockdown than tourism and recreation students and non-physical culture students;*


**H5**:
*Physical education students, regardless of gender, devote more attention per week during post-lockdown to intensive physical activity than tourism and recreation students and non-physical culture students.*


In the course of statistical analysis, the absence of normal distributions was clearly established, which gave rise to the use of non-parametric tests of significance of differences. In order to compare the groups of students of physical education, tourism and recreation, and non-physical education majors, a Mann–Whitney U test was used. A comparison of each of the aforementioned student groups, considering remote and traditional teaching, was conducted using the Wilcoxon test. On the other hand, a Kruskal–Wallis ANOVA test was used to compare the differences in physical activity during remote and traditional education between the studied groups of physical education students, tourism and recreation students, and non-physical education students. A level of statistical significance not exceeding 5% random error (*p* < 0.05) was assumed in all calculations. A Bonferroni correction was applied to avoid FWER (family-wise error rate), but the statistical significance of all differences was maintained at *p* < 0.05.

## 3. Results

Hypothesis 1 states that the level of physical activity calculated in METs in the groups of physical education students, tourism and recreation students and non-physical education students is higher during traditional teaching compared to the level of physical activity observed during remote teaching. The overall results are presented in Table 1 and Table 2.

Taking into account the different levels of effort intensity, it can be seen from the arithmetic means that PE students practiced moderate effort to a higher extent in lockdown (almost twice as much intensive and light), while this effort distribution in TR was similar, but much lower, and in the other majors, light activity dominated, while intensive activity was much lower. After lockdown, PE increased its intensive activity, while TR decreased, towards light activity—while the other faculties maintained similar changes to PE but remained at a lower level.

An analysis of the time spent on physical activity shows that, in lockdown, moderate exertion dominated in PE (most minutes spent), while light exertion took up more time in TR and other courses. Post lockdown in PE, the time of intensive activity increased, and the time of the other activity ranges decreased. In TR, the increase in time was proportional in all intensities. For the other directions, an increase in time was observed for intensive and moderate intensities, while it decreased for light exertion.

The results of the analysis verifying Hypothesis 1 in more detail are presented in Table 3 and Table 4.

For the PE group, the hypothesis (counted in MET) is false because the activity level counted in total MET is lower post lockdown than before (i.e., the difference is statistically significant, but the direction is the opposite of what was assumed). For TR and other fields of study, this part of the hypothesis is confirmed.

In terms of effort time measured in minutes, a difference in PE is evident (but the opposite of that assumed, so the hypothesis cannot be accepted), while the TR and other directions show no significant difference in time spent on physical activity, despite the concordant direction (the increase in time is too small relative to the spread to infer a significant increase).

Results verifying Hypothesis 2, which assumes that, during both remote and traditional teaching, the higher levels of total physical activity calculated in METs and minutes from physical education students compared to tourism and recreation students are presented in Table 5.

This hypothesis was only confirmed for the lockdown period.

The results verifying Hypothesis 3, assuming that, during both distance and traditional teaching, higher levels of total physical activity calculated in METs and minutes are shown by tourism and recreation students compared to non-physical culture students, are presented in Table 6.

Hypothesis 3 was verified positively.

The results verifying Hypothesis 4, assuming that physical education students, regardless of gender, devote more attention per week to intensive physical activity during lockdown than tourism and recreation students and non-physical culture students, are presented in Table 7 (MET) and Table 8 (minutes).

The hypothesis was fully confirmed.

The results verifying Hypothesis 5, assuming that physical education students, regardless of gender, devote more attention per week to intensive physical activity during post-lockdown than tourism and recreation students and non-physical culture students, are presented in Table 9 (MET) and Table 10 (minutes).

This hypothesis has also been fully confirmed.

## 4. Discussion

The environment, the second of the analyzed health determinants, is a very broad concept [23], and one of its proposed characteristics is the distinction between abiotic and biotic factors [24]. The former group is represented by the non-living components often referred to as the biotope and the latter by the living part of the ecosystem, known as the biocenosis. With reference to the rampant global health threat of the COVID-19 pandemic in recent months, it is very difficult to indisputably assign viruses to either of these two groups. Recently, however, concepts defending the hypothesis of viruses as animate elements have emerged [25]. Although they cannot replicate or carry out independent metabolic changes, the fact that they use the genetic code makes it difficult to deny their evolutionary relationship with the animate world [26]. From an ecological point of view, the relationship between viruses and humans has the character of a coaction in which the virus is the parasite and the human is the host [27]. The lack of pharmacological support in viral infection forces the emergence of adaptive intrinsic processes, stimulated largely by physical activity [28].

Undoubtedly, physical education students are the most engaged in physical activity compared to the other study groups. This applies to both MET values and time spent on weekly physical activity. This regularity is especially true for high physical activity. In general, it was noted that the more physical activity in the study program, the more engagement in physical activity. Thus, PE students are more physically active than TR students. In contrast, TR students are more physically active than NPC students. The advantage of TR students over NPC students was evident both during and after the lockdown period. However, an interesting observation is that there was no statistically significant difference between the physical activity levels of PE students and TR students in the post-lockdown period. Meanwhile, such a difference in favor of PE students was evident during the lockdown period. A likely explanation for this result is the significantly higher self-discipline among PE students. During the lockdown period, there was a lack of full didactic control over the exercising students. Students wishing to remain active and therefore physically fit despite this were more interested in physical exercise. As can be seen, this group was clearly PE students. This may be due to greater health awareness and, more specifically, a better recognition of physical activity as an element of a healthy lifestyle and also a condition for health. The study of physical education, which places particular emphasis on these issues in the Polish educational base, can provide support for positive health behavior.

Previous studies seem to confirm the key role of physical activity as an element of human activity during COVID-19. Evidence suggests that there is a positive association between NCD risk and physical inactivity [29,30]. To this end, many government agencies have developed guidelines for physical activity, not only as a strategy to prevent chronic disease but also in reference to its resulting psychological benefits [31,32]. A recent multicenter study found that COVID-19-enforced home isolation increased the percentage of physically inactive individuals by up to 15% [33]. Other findings from studies completed during the pandemic also argue for reduced physical activity during confinement, although the increased tendency for physical inactivity in this case is associated with adverse weather conditions as a reason for remaining indoors [30]. In this study, participants reduced the number of days with a minimum 10 min walk from 4.58 ± 1.57 days to 2.89 ± 1.40 days per week for women and from 4.38 ± 1.25 days to 2.67 ± 1.21 days per week for men. For both sexes, this resulted in a combined spectacular reduction in activity of about 40%. At the same time, walking time among women decreased from 56.9 ± 13.0 min to 27.2 ± 12.2 min and for men from 62.1 ± 10.6 min to 29.9 ± 13.5 min. Accordingly, weekly walking energy expenditure decreased from 868 ± 364 METs/min/week to 266 ± 188 METs/min/week for women and from 896 ± 301 METs/min/week for men to 273 ± 192 METs/min/week [30].

Since walking is a natural habit of human beings, this reduction in its volume convinces us that the so-called homo sedentarius has been created in recent times [34]. According to the authors of an international online study, the results of which were published in April 2020, the time spent walking was reduced by 35%, while in home isolation translated into 43% lower METs [33]. Another study found that energy expenditure due to home isolation decreased from 520 ± 372 METs/min/week to 238 ± 205 METs/min/week in women and from 663 ± 320 METs/min/week to 323 ± 187 METs/min/week in men [35]. The results of a Canadian adult study argue for the important role of exercise habits developed in the pre-pandemic period. They indicate that as many as 40.5% of previously inactive individuals and only 22.4% of previously active individuals became less active as a result of the pandemic. In comparison, 33% of previously inactive and 40.3% of previously active individuals became more active during the pandemic [36]. This allows us to accept with greater understanding the interpretation of the negatively verified first hypothesis in our study regarding the comparison of the remote and traditional activity of physical education students.

Some of the results of studies of the physical activity of university youth and concerning the pandemic period are quite optimistic. They speak, among others, of 43% participation in physical activity at the level of more than 150 min per week for Ukrainian students [37] and 56% of Chinese university students engaged in such activity at at least a moderate level [38]. However, as comparisons indicate, this is a significantly weaker result in relation to the pre-pandemic period in Ukraine, which is consistent with the results of some parallel studies [39,40]. Forced isolation due to COVID-19 also reduced the time spent on physical activity, while increasing sitting time and decreasing life satisfaction among university students in Qatar [41]. Deficits in mental space in conjunction with reduced physical activity, moreover, seem to be characteristic not only of the larger number of students surveyed during the pandemic period [42,43] but also of older subjects [44,45,46]. A harbinger of positive changes occurring as the pandemic lengthens is the gradual increase in physical activity and the reduction in time spent in sedentary behaviors [47].

## 5. Conclusions

The constraints associated with the spread of the COVID-19 pandemic and the resulting effects in the physical and mental health space may have important implications for post-lockdown, as well as post-COVID-19, health behaviors, including the extent of physical activity undertaken. Undoubtedly, this activity can be a key factor supporting the biological tolerance of the system and, connected with it, mitigating the negative individual and social effects of this period. Therefore, it should be included in health and prevention strategies from the local through the regional to the global level. In light of the presented results, the authors indicate the need to prevent sedentary habits, especially among representatives of the young generation with little or no educational connection to the issues of broadly defined physical culture. Neglect of physical activity at this stage of life may herald growing health problems in later years, becoming not only an individual problem of individuals but also, due to the massiveness of this phenomenon, a factor that reduces the quality of state health care for the citizen. The results of the presented research encourage the modification of the system of education in the area of the physical education of students, both in traditional and remote form. Especially in the latter case, it is worthwhile to plan the process of the transmission of didactic content more precisely and to improve the methods of its verification. This need becomes more and more important from the perspective of more and more frequent contacts between teachers and students in the online format and taking into account the likelihood of forced social isolation in the future.

## Figures and Tables

**Table 1 ijerph-19-11663-t001:** MET scores and time spent in physical activity during the lockdown period.

Variable	Field of Study	N	Mean	Min	Max	SD
Lockdown total physical activity (MET)	PE	73	5321.973	739	13,572	3336.069
Lockdown high physical activity (MET)	PE	73	1374.11	0	8000	1355.176
Lockdown moderate physical activity (MET)	PE	73	2567.123	0	9000	2083.898
Lockdown low physical activity (MET)	PE	73	1314.986	0	4158	1125.844
Lockdown total physical activity (min)	PE	73	1212.205	224	3180	784.126
Lockdown high physical activity (min)	PE	73	175.507	0	1000	174.423
Lockdown moderate physical activity (min)	PE	73	637.945	0	2250	524.304
Lockdown low physical activity (min)	PE	73	398.479	0	1260	341.165
Lockdown total physical activity (MET)	TR	65	2916.738	0	11,970	2330.85
Lockdown high physical activity (MET)	TR	65	847.077	0	7200	1292.601
Lockdown moderate physical activity (MET)	TR	65	1013.246	0	5170	1130.337
Lockdown low physical activity (MET)	TR	65	923.431	0	2772	868.999
Lockdown total physical activity (min)	TR	65	694.231	0	2145	507.355
Lockdown high physical activity (min)	TR	65	105.908	0	900	161.565
Lockdown moderate physical activity (min)	TR	65	257.615	0	1293	278.317
Lockdown low physical activity (min)	TR	65	303.231	0	1260	292.175
Lockdown total physical activity (MET)	NPC	62	1940.758	46	4659	1349.568
Lockdown high physical activity (MET)	NPC	62	353.79	0	2400	498.186
Lockdown moderate physical activity (MET)	NPC	62	729.032	0	3360	862.613
Lockdown low physical activity (MET)	NPC	62	845.194	0	3168	668.139
Lockdown total physical activity (min)	NPC	62	491.887	14	1440	341.172
Lockdown high physical activity (min)	NPC	62	52.452	0	420	82.89
Lockdown moderate physical activity (min)	NPC	62	182.323	0	840	215.61
Lockdown low physical activity (min)	NPC	62	256.145	0	960	202.452

Abbreviations: MET—metabolic equivalent of work. min—minutes. PE—physical education. TR—tourism and recreation. NPC—non-physical culture specialty.

**Table 2 ijerph-19-11663-t002:** MET scores and time spent in physical activity during the post-lockdown period.

Variable	Field of Study	N	Mean	Min	Max	SD
Post-lockdown total physical activity (MET)	PE	73	4343.329	318	13,314	2730.84
Post-lockdown high physical activity (MET)	PE	73	1692.603	0	5040	1371.89
Post-lockdown moderate physical activity (MET)	PE	73	1640.548	0	6900	1526.764
Post-lockdown low physical activity (MET)	PE	73	936.205	0	4158	839.422
Post-lockdown total physical activity (min)	PE	73	893.708	90	2870	600.182
Post-lockdown high physical activity (min)	PE	73	205.753	0	630	167.343
Post-lockdown moderate physical activity (min)	PE	73	401.795	0	1725	383.614
Post-lockdown low physical activity (min)	PE	73	286.575	0	1260	257.106
Post-lockdown total physical activity (MET)	TR	65	3584.8	198	11,970	2505.552
Post-lockdown high physical activity (MET)	TR	65	984.462	0	7200	1282.69
Post-lockdown moderate physical activity (MET)	TR	65	1366	0	4440	1140.622
Post-lockdown low physical activity (MET)	TR	65	1224.031	66	8316	1268.689
Post-lockdown total physical activity (min)	TR	65	797.092	60	2730	529.769
Post-lockdown high physical activity (min)	TR	65	123.077	0	900	160.326
Post-lockdown moderate physical activity (min)	TR	65	330.477	0	1110	276.971
Post-lockdown low physical activity (min)	TR	65	360.615	20	2520	380.874
Post-lockdown total physical activity (MET)	NPC	62	2819.468	231	18,342	2810.139
Post-lockdown high physical activity (MET)	NPC	62	910.855	0	10,080	1432.922
Post-lockdown moderate physical activity (MET)	NPC	62	1224.677	0	9200	1580.466
Post-lockdown low physical activity (MET)	NPC	62	685.419	0	2772	543.912
Post-lockdown total physical activity (min)	NPC	62	633.919	70	3472	561.812
Post-lockdown high physical activity (min)	NPC	62	119.968	0	1260	181.054
Post-lockdown moderate physical activity (min)	NPC	62	306.242	0	2300	395.042
Post-lockdown low physical activity (min)	NPC	62	207.71	0	840	164.819

**Table 3 ijerph-19-11663-t003:** Comparison of physical activity (MET) of students in MET in the lockdown and post-lockdown period (Wilcoxon’s signed-rank test).

Variable Pair	Field of Study	N	T	Z	*p*
Total lockdown (MET) vs. total post-lockdown (MET)	PE	62	**523.500**	**3.176**	**0.001**
Total lockdown (MET) vs. total post-lockdown (MET)	TR	61	**627.** **500**	**2.284**	**0** **.022**
Total lockdown (MET) vs. total post-lockdown (MET)	NPC	56	**457.000**	**2.781**	**0.005**

Results in bold are statistically significant.

**Table 4 ijerph-19-11663-t004:** Comparison of the physical activity (min) of students in MET in the lockdown and post-lockdown period (Wilcoxon’s signed-rank test).

Variable Pair	Field of Study	N	T	Z	*p*
Total lockdown (min) vs. total post-lockdown (min)	PE	60	**351.500**	**4.148**	**0.000**
Total lockdown (min) vs. total post-lockdown (min)	TR	61	778.000	1.203	0.228
Total lockdown (min) vs. total post-lockdown (min)	NPC	56	581.500	1.766	0.077

Results in bold are statistically significant.

**Table 5 ijerph-19-11663-t005:** Comparison of the amount of physical activity of PE and TR students in the lockdown and post-lockdown period (U Mann–Whitney test).

Variable	PE Sum of Rank	TR Sum of Rank	U	Z	*p*
Total lockdown (MET)	6214.000	3377.000	1232.000	**4.862**	**0.000**
Total lockdown (min)	6117.000	3474.000	1329.000	**4.448**	**0.000**
Total post-lockdown (MET)	5493.500	4097.500	1952.500	1.789	0.073
Total post-lockdown (min)	5294.500	4296.500	2151.500	2.940	0.346

Results in bold are statistically significant.

**Table 6 ijerph-19-11663-t006:** Comparison of the amount of physical activity of TR and NPC students in the. Lockdown and post-lockdown period (U Mann–Whitney test).

Variable	PE Sum of Rank	TR Sum of Rank	U	Z	*p*
Total lockdown (MET)	4650.000	3478.000	1525.000	**2.360**	**0.018**
Total lockdown (min)	4588.000	3540.000	1587.000	**2.061**	**0.039**
Total post-lockdown (MET)	4644.500	3483.500	1530.500	**2.334**	**0** **.019**
Total post-lockdown (min)	4631.500	3496.500	1543.500	**2.271**	**0** **.023**

Results in bold are statistically significant.

**Table 7 ijerph-19-11663-t007:** Comparison of the high physical activity measured in MET units during the lockdown period.

Field of Study	Kruskal–Wallis One-Way Analysis of Variance by Ranks: H (2, N = 200) = 40.50642 *p* = 0.0000			
	Code	N	Sum of Rank	Mean Rank
PE	1	73	9585.000	131.3014
TR	2	65	6257.500	96.2692
NPC	3	62	4257.500	68.6694

**Table 8 ijerph-19-11663-t008:** Comparison of time spent on high physical activity during the lockdown period.

Field of Study	Kruskal–Wallis One-Way Analysis of Variance by Ranks: H (2, N = 200) = 41.29383 *p* = 0.0000			
	Code	N	Sum of Rank	Mean Rank
PE	1	73	9754.500	133.6233
TR	2	65	5917.000	91.0308
NPC	3	62	4428.500	71.4274

**Table 9 ijerph-19-11663-t009:** Comparison of high physical activity measured in MET units during the post-lockdown period.

Field of Study	Kruskal–Wallis One-Way Analysis of Variance by Ranks: H (2, N = 200) = 19.33180 *p* = 0.0001			
	Code	N	Sum of Rank	Mean Rank
PE	1	73	9039.500	123.8288
TR	2	65	5878.000	90.4308
NPC	3	62	5182.500	83.5887

**Table 10 ijerph-19-11663-t010:** Comparison of time spent on high physical activity during the post-lockdown period.

Field of Study	Kruskal-Wallis One-Way Analysis of Variance by Ranks: H (2, N = 200) = 10.50672 *p* = 0.0052			
	Code	N	Sum of Rank	Mean Rank
PE	1	73	8265.000	113.2192
TR	2	65	6783.000	104.3538
NPC	3	62	5052.000	81.4839

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
