# Peer review of "Physical Culture and Sports as an Educational Basis of Students’ Healthy Physical Activities during and Post-Lockdown COVID-19 Restrictions"

_ijerph, 2022, doi:10.3390/ijerph191811663_

Round 1

Reviewer 1 Report

Title:  Physical culture and sports as an educational basis of students’ healthy physical activities during and post-lockdown COVID-19 restrictions

Thanks for inviting me to review this manuscript. This study examined the level of physical activity in groups of students with different levels of educational engagement in physical activity including students of physical education (n - 91), tourism and recreation (n - 16), and students of a non-physical education major (n - 94). I have some concerns about the rational and methods of this study.

Introduction

-          In general, I don’t the see how the Introduction (i.e., literature review) leads to the need of this study. For example, the introduction even doesn’t mention any key words in the study title such as “physical culture”, “sports”, “educational basis”, and “COVID-19 restriction”.

-          It says that “Recently, many international research communities recommend IPAQ.”   The IPAQ has been widely used for at least 20 years so it is not a recent recommendation at all.

-          This study compared between different groups of students, but the introduction doesn’t show any background and rational why the 3 groups are examined. Also, no rationale why the realities of distance and traditional education were taken into account.

Methods

-          The sample was not clearly described such as location, college students, age range, sex, race/ethnicity, etc.

-          Have the IPAQ data been adjusted in the data cleaning process based on Guidelines for Data Processing and Analysis of the IPAQ (https://www.researchgate.net/file.PostFileLoader.html?id=5641f4c36143250eac8b45b7&assetKey=AS%3A294237418606593%401447163075131)? Given the self-reported data, some outliers may need to be excluded; for example, “All cases in which the sum total of all Walking, Moderate and Vigorous time variables is greater than 960 minutes (16 hours) should be excluded from the analysis.”

-          Did the authors transform the data to check the possibility of using parametric analyses given the absence of normal distributions?

-          Have the analyses been adjusted for multiple comparisons using Bonferroni correction, for example?

-          Why was only MET-min measure reported? I think PA time (hours or minutes) should be more straightforward.

-          Also, the analyses should be conducted by PA types like walking, moderate PA, and vigorous PA as the people’s activity habits may shift between the types during the pandemic. The overall MET-time may not catch the changes effectively.

Results

-          Although non-parametric analyses were used, it still worth providing a table to show means of three groups during and post-lockdown COVID-19 restrictions so readers have some sense about the magnitudes of the values.

-          To take advantage of the data with both during and post-lockdown COVID data, the analysis to compare the changes (between during and post) across three groups should be conducted. A trend figure to show the change between during and post by group will be helpful to visualize the data.

-          Kruskal-Wallis ANVOA test was used but why the analysis was stratified with remote learning (Table 4) and Traditional Leaning (Table 5) separately?  

Discussion

-          I didn’t notice any discussion related to the “physical culture” and “educational basis”.  And I have difficulty in understanding the links between the measures and the title of the study.

-          Also, I don’t think some discussions are closely related to this study, such as the paragraph “Previous studies seem to confirm the key role of physical activity as a health determinant.”

-          I don’t think walking alone can interpret the hypothesis 1 in the paragraph (lines 236  - 250) since the authors used overall MET-min measure rather than by intensity specific PA.  

-          In the Conclusion, it says, “The results of the presented research encourage modification of the system of education in the area of physical education of students, both in traditional and remote form. Especially in the latter case, it is worthwhile to plan the process of transmission of didactic content more precisely and to improve the methods of its verification.”  No evidence was given that there is a difference between the studies three groups in terms of PE because readers cannot assume that PE students have better knowledge and skills in PE compared to other groups of students.

Author Response

Dear Reviewer
We are very sorry for the long wait to improve the manuscript. During the summer we had difficult contact. In the meantime, your suggestions required a very significant improvement of the article. 
Below are our responses to your questions and concerns.

Introduction

1. In general, I don’t the see how the Introduction (i.e., literature review) leads to the need of this study. For example, the introduction even doesn’t mention any key words in the study title such as “physical culture”, “sports”, “educational basis”, and “COVID-19 restriction”.

In the introduction, several issues vaguely related to the title of the manuscript were dropped and moved to the Discussion. On the other hand, the topic of the educational basis in the Polish reality of university functioning has been placed in the second paragraph of the introduction (lines 35 - 55).

2.  It says that “Recently, many international research communities recommend IPAQ.”   The IPAQ has been widely used for at least 20 years so it is not a recent recommendation at all.

This amendment has been included in the text (line 81).

3. This study compared between different groups of students, but the introduction doesn’t show any background and rational why the 3 groups are examined. Also, no rationale why the realities of distance and traditional education were taken into account.

The authors have tried to address this point in the introduction (lines 51 - 55).

Methods

4. The sample was not clearly described such as location, college students, age range, sex, race/ethnicity, etc.

A detailed description of the study group can be found in the revised version of the manuscript, lines 99 - 106.

5. Have the IPAQ data been adjusted in the data cleaning process based on Guidelines for Data Processing and Analysis of the IPAQ (https://www.researchgate.net/file.PostFileLoader.html?id=5641f4c36143250eac8b45b7&assetKey=AS%3A294237418606593%401447163075131)? Given the self-reported data, some outliers may need to be excluded; for example, “All cases in which the sum total of all Walking, Moderate and Vigorous time variables is greater than 960 minutes (16 hours) should be excluded from the analysis.”

This very important information resulted in the rejection of quite a large number of respondents, and deficiencies in the questionnaires regarding the detailed description of high, moderate and low activities reduced the number of students surveyed from 293 to 200.

6. Did the authors transform the data to check the possibility of using parametric analyses given the absence of normal distributions?

After transforming the data and checking the normality of the distributions, the authors applied non-parametric tests.

7. Have the analyses been adjusted for multiple comparisons using Bonferroni correction, for example?

The authors have referred to Bonferroni's correction in the text (lines 149 - 151).

8. Why was only MET-min measure reported? I think PA time (hours or minutes) should be more straightforward.

In addition to the MET-min values, the revised version of the manuscript also made calculations for physical activity time in minutes.

9. Also, the analyses should be conducted by PA types like walking, moderate PA, and vigorous PA as the people’s activity habits may shift between the types during the pandemic. The overall MET-time may not catch the changes effectively.

The revised version of the results presents all types of physical activity. Consequently, the number and content of the research hypotheses have also been changed.

Results

10. Although non-parametric analyses were used, it still worth providing a table to show means of three groups during and post-lockdown COVID-19 restrictions so readers have some sense about the magnitudes of the values.

These calculations can be found in Tables 1 and 2.

11. To take advantage of the data with both during and post-lockdown COVID data, the analysis to compare the changes (between during and post) across three groups should be conducted. A trend figure to show the change between during and post by group will be helpful to visualize the data.

In the revised version of the manuscript, the authors have attempted to present summaries from lockdown and post-lockdown times. They are presented in Tables 3-6.

12. Kruskal-Wallis ANVOA test was used but why the analysis was stratified with remote learning (Table 4) and Traditional Leaning (Table 5) separately? 

The Kruskal-Wallis-ANOVA test was included in the revised version of the manuscript and was helpful in the verification of hypothesis 4 and 5 (tables 7-10). Alternatively, we request further comments on this issue.

Discussion

13. I didn’t notice any discussion related to the “physical culture” and “educational basis”.  And I have difficulty in understanding the links between the measures and the title of the study.

The authors introduced this element of discussion in lines 35 - 55, and additionally returned to the educational base in lines 275 - 279.

14. Also, I don’t think some discussions are closely related to this study, such as the paragraph “Previous studies seem to confirm the key role of physical activity as a health determinant.”

The authors resigned this overly general reference to the research issue.

15.  I don’t think walking alone can interpret the hypothesis 1 in the paragraph (lines 236  - 250) since the authors used overall MET-min measure rather than by intensity specific PA. 

The authors have referred to this in previous statements.

16. In the Conclusion, it says, “The results of the presented research encourage modification of the system of education in the area of physical education of students, both in traditional and remote form. Especially in the latter case, it is worthwhile to plan the process of transmission of didactic content more precisely and to improve the methods of its verification.”  No evidence was given that there is a difference between the studies three groups in terms of PE because readers cannot assume that PE students have better knowledge and skills in PE compared to other groups of students.

The revised Results section demonstrates the differences between the representatives of the 3 research groups and a brief reference to the study programme is provided in lines 261 - 279.

Reviewer 2 Report

The sample is described very briefly, there are no socio-demographic data, it is necessary to prove the typicality and representativeness of the sample. On the example of this sample, is it possible to formulate conclusions about all students who study in these directions, which are the grounds for this?

A clear description of the research procedure is necessary, how and when it was organized, how the survey was conducted, etc. In what way was the study organized to compare the differences in physical activity during remote and traditional education?

The introduction does not fully correspond to the topic of the article, it requires revision, perhaps greater emphasis on educational programs in physical education and sports

Taking into account the presence of differences in the physical activity of men and women, which the authors also state in the discussion (lines 227-242), it is worth presenting own results taking into account the gender of the respondent

The discussion should more clearly emphasize the uniqueness and strengths of this study

Author Response

Dear Reviewer
We are very sorry for the long wait to improve the manuscript. During the summer we had difficult contact. In the meantime, your suggestions required a very significant improvement of the article. 
Below are our responses to your questions and concerns.

1. The sample is described very briefly, there are no socio-demographic data, it is necessary to prove the typicality and representativeness of the sample. On the example of this sample, is it possible to formulate conclusions about all students who study in these directions, which are the grounds for this?

The results presented require additional verification with a larger sample. Due to the constraints of the pandemic period and the possibility of directly reaching part of the student population, this research sample comprises 200 people. The characteristics of the research group can be found in lines 99 - 106.

2. A clear description of the research procedure is necessary, how and when it was organized, how the survey was conducted, etc. In what way was the study organized to compare the differences in physical activity during remote and traditional education?

The authors have tried to address this comment in the revised version of the manuscript (lines 117 - 120).

3. The introduction does not fully correspond to the topic of the article, it requires revision, perhaps greater emphasis on educational programs in physical education and sports

In the introduction, several issues vaguely related to the title of the manuscript were dropped and moved to the Discussion. On the other hand, the topic of the educational basis in the Polish reality of university functioning has been placed in the second paragraph of the introduction (lines 35 - 55).

4. Taking into account the presence of differences in the physical activity of men and women, which the authors also state in the discussion (lines 227-242), it is worth presenting own results taking into account the gender of the respondent.

This interesting proposal has been considered by the authors and is to become part of a separate research study.

5. The discussion should more clearly emphasize the uniqueness and strengths of this study.

The authors tried to take this suggestion into account by improving several sections of the test in the Discussion.

Reviewer 3 Report

Abstract:

The results should respond to the main objective of the study. What were the levels of physical activity? And in the results should appear values.

“while taking into account the realities of remote and traditional education” – I did not understand. It was not “during and post-lockdown COVID-19 restrictions”?

You cannot conclude this because you did not compare with students from other courses and at the conclusion you must answer the main objective.

Introduction:

The authors do not substantiate the main research problem. Write content that will not be evaluated, confusing the reader and not addressing the problem. Why study physical activity course students specifically during COVID? What is the relationship? What led you to do this research? This should be well-founded in your introduction. They are entire paragraphs with information that could be included, but could be perfectly unnecessary and/or well summarized because it does not support the problem.

If you choose to write hypotheses, these must be made in the methodology

Methods

They did not mention the type of study, ethical considerations, the location of the study, the population and sample calculation, the validation of the measurement instrument, the study period, when the instrument was applied (during and post-covid) , whoever applied it, should explain the measurement instrument in more detail, such as how the calculation is done, which test was used to test the normality of the data?

Didn't mention any of that in the introduction – “A comparison of each of the aforementioned student groups considering remote and traditional teaching”

“considering remote and traditional teaching” - this has to be substantiated

I didn't understand if the physical activity is carried out in classes, whether online or in person, or if physical activities carried out outside the school environment are accounted for

Results

during traditional teaching compared during remote teaching. - Does that mean during and after lockdown? Each country had its restrictions. For example, in my country, in the 2nd confinement, practical classes were allowed with the use of a mask, so it is important to substantiate the problem in the Introduction referring to the measures adopted in the country where the study was carried out, the period of data collection.

The results do not present hypotheses, they simply present the results.

I did not correct the discussion as there are many aspects to be clarified.

Author Response

Dear Reviewer
We are very sorry for the long wait to improve the manuscript. During the summer we had difficult contact. In the meantime, your suggestions required a very significant improvement of the article. 
Below are our responses to your questions and concerns.

1. The results should respond to the main objective of the study. What were the levels of physical activity? And in the results should appear values.

“while taking into account the realities of remote and traditional education” – I did not understand. It was not “during and post-lockdown COVID-19 restrictions”?

You cannot conclude this because you did not compare with students from other courses and at the conclusion you must answer the main objective.

The aim of the study was significantly modified after comments from reviewers. Together with the description, it can be found in lines 87 - 97.

Introduction

2. The authors do not substantiate the main research problem. Write content that will not be evaluated, confusing the reader and not addressing the problem. Why study physical activity course students specifically during COVID? What is the relationship? What led you to do this research? This should be well-founded in your introduction. They are entire paragraphs with information that could be included, but could be perfectly unnecessary and/or well summarized because it does not support the problem. 

In the introduction, several issues vaguely related to the title of the manuscript were dropped and moved to the Discussion. On the other hand, the topic of the educational basis in the Polish reality of university functioning has been placed in the second paragraph of the introduction (lines 35 - 55).

3. If you choose to write hypotheses, these must be made in the methodology.

The authors have corrected this error in the structure of the manuscript's content.

Methods

4. They did not mention the type of study, ethical considerations, the location of the study, the population and sample calculation, the validation of the measurement instrument, the study period, when the instrument was applied (during and post-covid) , whoever applied it, should explain the measurement instrument in more detail, such as how the calculation is done, which test was used to test the normality of the data?

The measurement tool was used previously on a representative number of respondents in the Polish authors' work: Biernat, E.; Stupnicki, R.; Gajewski, A. K. MiÄ™dzynarodowy Kwestionariusz AktywnoÅ›ci Fizycznej (IPAQ) Wersja Polska [International Physical Activity Questionnaire (IPAQ) Polish Version]. Wych. Fiz. Sport 2007, 51(1), 47–54.

The time of the research (line 118 - 120) and the ethical conditions of the research - Institutional Review Board Statement (line 352 - 358) - have been added to the text.

5. Didn't mention any of that in the introduction – “A comparison of each of the aforementioned student groups considering remote and traditional teaching”

“considering remote and traditional teaching” - this has to be substantiated

I didn't understand if the physical activity is carried out in classes, whether online or in person, or if physical activities carried out outside the school environment are accounted for.

In describing the purpose of the study, the authors sought to clarify these concerns (lines 87 - 97).

Results

6. During traditional teaching compared during remote teaching. - Does that mean during and after lockdown? Each country had its restrictions. For example, in my country, in the 2nd confinement, practical classes were allowed with the use of a mask, so it is important to substantiate the problem in the Introduction referring to the measures adopted in the country where the study was carried out, the period of data collection.

In the introductory part of the manuscript, the authors tried to clarify these doubts (lines 35 - 55).

7. The results do not present hypotheses, they simply present the results.

The authors have almost completely revised the chapter presenting the results of the study.

8. I did not correct the discussion as there are many aspects to be clarified.

The authors look forward to the Reviewer's comments on the Discussion.

Round 2

Reviewer 1 Report

NA

Author Response

Dear Editor

Thank you for your prompt response to the numerous changes we made to the structure of the manuscript.

We appreciate your contribution to improving the quality of the presented text.

Kind regards,

Marcin Pasek

Reviewer 2 Report

Dear Authors,

thank you for your reply and consideration of my recommendations. I look forward to further research along these lines. I wish you success in your future endeavors.

Kind regards,

Author Response

Dear Reviewer

Thank you for your quick response to the numerous changes we made to the structure of the manuscript.

We appreciate your contribution to improving the quality of the text presented.

We have made the spelling corrections you suggested in the text using Track Changes.

Kind regards,

Marcin Pasek